# Deep Neural Networks for Quality Assurance of Image Registration

**S Bannister**[1i]**, D Page**[1ii]**, T Standen**[1]**, A Dunne**[1]**, J Rawling**[1]**, C J Birch-Sykes**[1]**,
M Z Wilson**[2]**, S Holloway**[2]**, J McClelland**[3]**, Y Peters**[1]

[1] *University of Manchester*

[2] *Department of Medical Physics and Biomedical Engineering, UCL, London, UK*

[3] *Centre for Medical Image Computing, UCL, UK*

## Abstract

Patient-specific quality assurance of image registrations is needed to enable their use in adaptive radiotherapy. An automated method of assessing the quality of a registration between head and neck CT scans was investigated. Ground truth organ contours were propagated to the deformed image from the floating image and then compared with contours on the reference image by calculating Dice similarity. A 2D convolutional neural network was designed to predict Dice coefficients based on the reference image and the displacement vector field of the registration. The network was trained using axial slices of head and neck CT images to predict the metric for the spinal canal, parotid glands and brainstem. The network was able to predict the Dice coefficient for unseen images. For the spinal canal, 95% of predictions were within 0.208 of the true value, with an average absolute difference of 0.0811. For the left parotid gland, 95% were within 0.270 of the true value, with an average absolute difference of 0.0987. This demonstrates that convolutional neural networks can be trained to effectively predict similarity metrics which can be used to assess the quality of an automatically produced registration.

## 1. Introduction

Image registration has potential clinical applications in adaptive radiotherapy (Veiga et al., 2015), but it must be accurate to ensure anatomy and dose field are mapped to the correct locations for the purposes of plan adaptation or dose accumulation. Currently, methods are available to assess the general performance of registration algorithms (Yeap et al., 2017), but if such algorithms are to be used clinically, manual checking is required for each patient. Previously, some work has been done to automatically provide quality assurance for the registrations of individual patient images. One method propagates a ground truth contour from one image, then back-propagates it onto a second image for validation, requiring contours on two images (Beasley et al., 2016). Another method uses a reverse classifier to predict contours for images for which ground truth contours are available, and the accuracy of these predictions is used to estimate the true accuracy of the registration (Valindria et al., 2017).

In this project, an alternative method for patient-specific quality assurance of image registration was investigated. This method involves designing a convolutional neural network (CNN) that takes the reference image of a registration and the corresponding displacement vector field, which describes the transformation between the reference and floating images,

as inputs. The network predicts similarity metrics between the transformed and original organ contours, producing quantitative measures of registration performance in the absence of labels on one of the images. This would allow for registrations with poor predicted similarity metrics to be flagged for manual review.This project focuses on inter-patient registrations, as more inter-patient data is available to train a network on. This could later be extended to intra-patient registrations, as this would be the scenario in a clinical setting.

## 2. Methodology

The data for the project was obtained from the Head-Neck Cetuximab dataset of The Cancer Imaging Archive. This data contained three dimensional scans of the head and neck region and, for most patients, contours of the spinal canal, brainstem and left and right parotid glands (Bosch et al., 2015). Approximately 2000 sets of inter-patient deformable registrations were created from an initial 94 patient scans. The corresponding displacement vector fields were also generated and contours were propagated from the reference image to the deformed image. As the project investigated the effectiveness of this method on two dimensional images, all images were divided into two dimensional slices that could be used as inputs for the network.

Table 1: The order and output dimensions of the layers in the CNN architecture used for this project. The convolutional layers had filters of dimension $3 \times 3$ and a $1 \times 1$ stride, with 32, 64 and 128 filters respectively and the max pooling layers each had $2 \times 2$ pooling windows and $2 \times 2$ stride. Training was done using stochastic gradient descent.

| Layer | Output dimensions |
| --- | --- |
| Input | $512 \times 512 \times 4$ |
| 2D Convolution | $510 \times 510 \times 32$ |
| Batch Normalisation | $510 \times 510 \times 32$ |
| 2D Max Pooling | $255 \times 255 \times 32$ |
| 2D Convolution | $253 \times 253 \times 64$ |
| Batch Normalisation | $253 \times 253 \times 64$ |
| 2D Max Pooling | $126 \times 126 \times 64$ |
| 2D Convolution | $124 \times 124 \times 128$ |
| Batch Normalisation | $124 \times 124 \times 128$ |
| Flatten | 1968128 |
| Dense | 32 |
| Batch Normalisation | 32 |
| Dense | 1 |

The volume-based Dice coefficient was calculated for the deformable and reference contours for the supervision used during training. The technique was investigated for spinal canals, brainstems and parotid glands, chosen due to their close proximity in the neck region, where accurate dosage delivery could greatly reduce radiotherapy-induced complications. The same network architecture, displayed in Table 1, was used for each with a mean squared error loss function and and a learning rate of 0.01. This network was purpose

built to predict Dice coefficients and optimized using spinal canal images. The behaviour of the network was investigated using $4 \times 4$ black box occlusion maps, illustrating which regions of the input images were most influential to the network output.

## 3. Results and Discussion

The technique was tested on spinal canals, brainstems and parotid glands. The predictions for the spinal canal and left parotid gland are depicted in Figure 1. For the spinal canal, training was conducted with 3619 slices and validation with 1552. The left parotid involved 793 slices for training and 341 for validation.

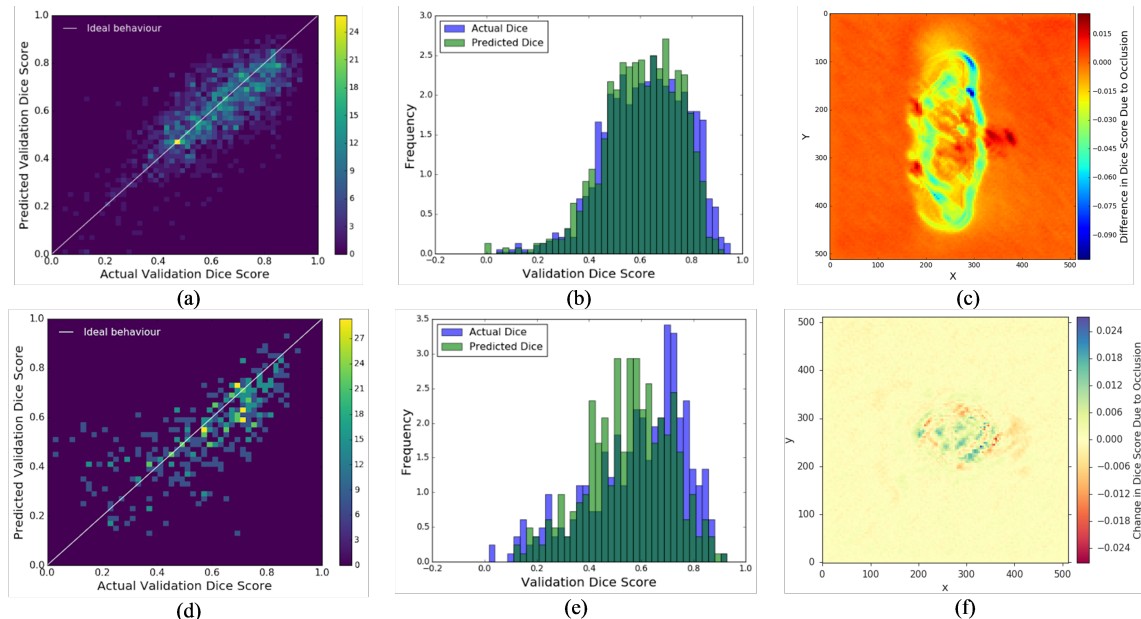

Figure 1: (a) A 2D histogram depicting the relationship between the predicted and true values of the Dice coefficient for the spinal canal validation data with a line of ideal behaviour. (b) A 1D histogram for the same spinal canal data. (c) An example of an occlusion map for the spinal canal. (d),(e) and (f) display the analogous results for the left parotid gland.

The absolute difference between the predicted and true Dice scores was calculated. For the validation data, the 95th percentile of this difference was 0.208 for the spinal canal and 0.270 for the left parotid. The mean difference was 0.0811 for the spinal canal and 0.0987 for the left parotid.

**Conclusion:** We have produced evidence that accurate predictions of Dice coefficients can be made by CNNs, demonstrating a potential solution to quality assurance concerns for automated image registration of patient scans.

## Acknowledgments

We would like to acknowledge Global Challenge Network+ in Advanced Radiotherapy for establishing this project.

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

## Notes

[i]sarah.bannister-2@student.manchester.ac.uk
[ii]denis.page@student.manchester.ac.uk

