# OpenReview forum: "Deep Neural Networks for Quality Assurance of Image Registration"
_MIDL.io/2019/Conference/Abstract — MIDL Abstract 2019_

### Official Review · AnonReviewer1 · 2019-04-30
**incomplete results**

**Rating:** 2
**Confidence:** 2

**Review:**

The authors proposed a method for image registration quality assurance by predicting similarity metrics between the transformed and original organ contours with a convolutional neural network. The authors showed that the proposed network is indeed able to accurately predict the similarity score (DICE coefficient), however further experiments are lacking to show the effectiveness of the method on quality assurance:
1) the results do not directly show how successfully the method can identify poor registration;
2) experiments with occlusions were also carried out, however the most influential regions on the output were not visualised.

---

### Official Review · AnonReviewer2 · 2019-05-01
**Interesting**

**Rating:** 3
**Confidence:** 3

**Review:**

The work aims to use a neural network to verify the registration of images. This seems like an interesting challenge. The work uses open data from the TCIA which will enable the work to have impact in developing and understanding the algorithm they are developing.

Some notes:
Having a flat layer of 1968128 units seems a bit large.
In the evaluation it does not indicate if the validation and test sets had unique patients.

---

### Decision · Program_Chairs · 2019-05-06
**Acceptance Decision**

Accept